# INFORMATION THEORETIC LOWER BOUNDS ON NEGATIVE LOG LIKELIHOOD

**Luis A. Lastras-Montaño**
IBM Research AI
Yorktown Heights, NY, 10598, USA
`lastrasl@us.ibm.com`

## ABSTRACT

In this article we use rate-distortion theory, a branch of information theory devoted to the problem of lossy compression, to shed light on an important problem in latent variable modeling of data: is there room to improve the model? One way to address this question is to find an upper bound on the probability (equivalently a lower bound on the negative log likelihood) that the model can assign to some data as one varies the prior and/or the likelihood function in a latent variable model. The core of our contribution is to formally show that the problem of optimizing priors in latent variable models is exactly an instance of the variational optimization problem that information theorists solve when computing rate-distortion functions, and then to use this to derive a lower bound on negative log likelihood. Moreover, we will show that if changing the prior can improve the log likelihood, then there is a way to change the likelihood function instead and attain the same log likelihood, and thus rate-distortion theory is of relevance to both optimizing priors as well as optimizing likelihood functions. We will experimentally argue for the usefulness of quantities derived from rate-distortion theory in latent variable modeling by applying them to a problem in image modeling.

## 1 INTRODUCTION

A statistician plans to use a latent variable model

$$p(x) = \int p(z)\ell(x|z)dz, \tag{1}$$

where $p(z)$ is known as the prior over the latent variables, and $\ell(x|z)$ is the likelihood of the data conditional on the latent variables; we will often use $(p(z), \ell(x|z))$ as a shorthand for the model. Frequently, both the prior and the likelihood are parametrized and the statistician's job is to find reasonable parametric families for both - an optimization algorithm then chooses the parameter within those families. The task of designing these parametric families can sometimes be time consuming - this is one of the key modeling tasks when one adopts the representation learning viewpoint in machine learning.

In this article we ask the question of how much $p(z)$ can be improved if one fixes $\ell(x|z)$ and viceversa, with the goal of equipping the statistician with tools to make decisions on where to invest her time. One way to answer whether $p(z)$ can be improved for fixed $\ell(x|z)$ is to drop the assumption that $p(z)$ must belong to a particular family and ask how a model could improve in an unrestricted setting. Mathematically, given data $\{x_1, \cdots, x_N\}$ the first problem we study is the following optimization problem: for a fixed $\ell(x|z)$,

$$\min_{p(z)} -\frac{1}{N} \sum_{i=1}^{N} \log \int p(z)\ell(x_i|z)dz \tag{2}$$

which as we will show, is also indirectly connected to the problem of determining if $\ell(x|z)$ can be improved for a given fixed $p(z)$. The quantity being optimized in (2) is called the average negative log likelihood of the data, and is used whenever one assumes that the data $\{x_1, \cdots, x_N\}$ have been

drawn statistically independently at random. Note that in this paper we are overloading the meaning of $p(z)$: in (1) it refers to prior in the "current" latent variable model in the statistician's hands, and in (2) and other similar equations, it refers to a prior that can be optimized. We believe that the meaning will be clear depending on the context.

Obviously, for any given $\ell(x|z), p(z)$, from the definition (1) we have the trivial upper bound

$$\min_{p(z)} -\frac{1}{N} \sum_{i=1}^{N} \log \int p(z)\ell(x_i|z)dz \leq -\frac{1}{N} \sum_{i=1}^{N} \log p(x_i) \tag{3}$$

Can we give a good *lower bound*? A lower bound could tell us how far we can improve the model by changing the prior. The answer turns out to be in the affirmative. In an important paper, Lindsay (1983) proved several facts about the problem of optimizing priors in latent variable models. In particular, he showed that

$$\min_{p(z)} -\frac{1}{N} \sum_{i=1}^{N} \log \int p(z)\ell(x_i|z)dz \geq -\frac{1}{N} \sum_{i=1}^{N} \log p(x_i) - \sup_z \log \left( \frac{1}{N} \sum_{i=1}^{N} \frac{\ell(x_i|z)}{p(x_i)} \right). \tag{4}$$

This result is very general - it holds for both discrete and continuous latent variable spaces, scalar or vector. It is also *sharp* - if you plug in the right prior, the upper and lower bounds match. It also has the advantage that the lower bound is written as a function of the trivial upper bound (3) - if someone proposes a latent variable model $p(x)$ which uses a likelihood function $\ell(x|z)$, the optimal negative log likelihood value when we optimize the prior is thus known to be within a gap of

$$\sup_z \log \left( \frac{1}{N} \sum_{i=1}^{N} \frac{\ell(x_i|z)}{p(x_i)} \right) \tag{5}$$

bits. The individual quantities under the $\sup$

$$c(z) \triangleq \frac{1}{N} \sum_{i=1}^{N} \frac{\ell(x_i|z)}{p(x_i)} \tag{6}$$

have a specific functional meaning: they can be regarded as multiplicative factors that tell you how to modify the prior to improve the log likelihood (see the Blahut-Arimoto algorithm (Blahut, 1972)):

$$p(z) \leftarrow p(z)c(z). \tag{7}$$

Lindsay (1983) derived his results with no apparent reference to earlier published work on rate-distortion theory, which is how information theorists study the problem of lossy compression Shannon (1959). There is no reason for why this connection could be reasonably made, as it is not immediately obvious that the problems are connected; However, the quantity $c(z)$ and the lower bound (4) in fact can be derived from ideas in Berger's book on rate-distortion theory (Berger, 1971); in fact Lindsey's classical result that the optimal prior in a latent variable model has finite support with size equal to the size of the training data can be, drawing the appropriate connection, seen as a result of a similar result in rate-distortion theory also contained in Berger (1971).

With time, the fundamental connection between the log likelihood optimization in latent variable modeling and the computation of rate-distortion functions became more clear. Although not explicitly mentioned in these terms, Rose (1998), Neal & Hinton (1998) restate the optimal log likelihood as a problem of minimizing the variational free energy of a certain statistical system; this expression is identical to the one that is optimized in rate-distortion theory. The Information Bottleneck method of Tishby et al. (1999) is a highly successful idea that exists in this boundary, having created a subfield of research that remains relevant nowadays (Tishby & Zaslavsky, 2015) (Shwartz-Ziv & Tishby, 2017). Slonim & Weiss (2002) showed a fundamental connection between maximum likelihood and the information bottleneck method: *"every fixed point of the IB-functional defines a fixed point of the (log) likelihood and vice versa"*. Banerjee et al. (2004) defined a rate-distortion problem where the output alphabet $\mathcal{Z}$ is finite and is jointly optimized with the test channel. By specializing to the case of where the distortion measure is a Bregman divergence, they showed the mathematical equivalence between this problem and that of maximum likelihood estimation where the likelihood function is an exponential family distribution derived from the Bregman divergence. Watanabe & Ikeda (2015) study a variation of the maximum likelihood estimation for latent variable

models where the maximum likelihood criterion is instead replaced with the entropic risk measure. The autoencoder concept extensively used in the neural network community is arguably directly motivated by the encoder/decoder concepts in lossy/lossless compression. Giraldo & Príncipe (2013) proposed using a matrix based expression motivated by rate-distortion ideas in order to train autoencoders while avoiding estimating mutual information directly. Recently, Alemi et al. (2018) exploited the $\beta$-VAE loss function of Higgins et al. (2017) to explicitly introduce a trade-off between rate and distortion in the latent variable modeling problem, where the notions of rate and distortion have similarities to those used in our article.

Latent Variable Modeling is undergoing an exciting moment as it is a form of *representation learning*, the latter having shown to be an important tool in reaching remarkable performance in difficult machine learning problems while simultaneously avoiding feature engineering. This prompted us to look deeper into rate-distortion theory as a tool for developing a theory of representation learning. What excites us is the possibility of using a large repertoire of tools, algorithms and theoretical results in lossy compression in meaningful ways in latent variable modeling; notably, we believe that beyond what we will state in this article, Shannon's famous lossy source coding theorem, the information theory of multiple senders and receivers, and the rich Shannon-style equalities and inequalities involving entropy and mutual information are all of fundamental relevance to the classical problem (1) and more complex variations of it.

The goal of this article is laying down a firm foundation that we can use to build towards the program above. We start by proving the fundamental equivalence between these two fields by using simple convexity arguments, avoiding the variational calculus arguments that had been used before. We then take an alternate path to proving (2) also involving simple convexity arguments inspired by earlier results in rate-distortion theory.

We then focus on what is a common problem - instead of improving a prior for a fixed likelihood function, improve the likelihood function for a fixed prior but for a fixed prior. Interestingly, rate-distortion theory still is relevant to this problem, although the question that we are able to answer with it is smaller in scope. Through a simple change of variable argument, we will argue that if the negative log likelihood can be improved by modifying the prior, exactly the same negative log likelihood can be attained by modifying the likelihood function instead. Thus if rate-distortion theory predicts that there is scope for improvement for a prior, the same holds for the likelihood function but conversely, while rate-distortion theory can precisely determine when it is that a prior can no longer be improved, the same cannot be said for the likelihood function.

Finally, we test whether the lower bound derived and the corresponding fundamental quantity $c(z)$ are useful in practice when making modeling decisions by applying these ideas to a problem in image modeling for which there have been several recent results involving Variational Autoencoders.

## 2 TECHNICAL PRELIMINARIES

In our treatment, we will use notation that is commonly used in information theory. If you have two distributions $P, Q$ the KL divergence from $P$ to $Q$ is denoted as $D_{\mathrm{KL}}(P\|Q)$ which is known to be nonnegative. For a discrete random variable $A$, we denote its entropy $H(A)$. If you have two random variables $A, B$, their mutual information is

$$I(A; B)$$

We assume that the data $\{x_1, \cdots, x_N\}$ comes from some arbitrary alphabet $\mathcal{X}$ which could be continuous or discrete. For example, it could be the set of all sentences of length up to a given number of words, the set of all real valued images of a given dimension, or the set of all real valued time series with a given number of samples.

Let $X$ be a random variable distributed uniformly over the training data $\{x_1, \cdots, x_N\}$. The entropy $H(X)$ of this random variable is $\log N$, and the fundamental lossless compression theorem of Shannon tells us that any compression technique for independently, identically distributed data samples following the law of $X$ must use at least $H(X)$ bits per sample. But what if one is willing to allow losses in the data reconstruction? This is where rate-distortion theory comes into play. Shannon introduced the idea of a reconstruction alphabet $\mathcal{Z}$ (which need not be equal to $\mathcal{X}$), and a *distortion function $d : \mathcal{X} \times \mathcal{Z} \to \mathbb{R}$* which is used to measure the cost of reproducing an element of $\mathcal{X}$ with an

element of $\mathcal{Z}$. He then introduced an auxiliary random variable $Z$, which is obtained by passing $X$ through a channel $Q_{Z|X}$, and defined the rate-distortion function as

$$R(D) = \min_{Q_{Z|X}: E_{X,z} d(X,Z) \leq D} I(X;Z) \tag{8}$$

Shannon's fundamental lossy source coding theorem in essence shows that $R(D)$ plays the same role as the entropy plays for lossless compression - it represents the minimum number of bits per sample needed to compress $X$ within a fidelity $D$, showing both necessity and sufficiency when the number of samples being compressed approaches infinity. This fundamental result though is *not needed* for this paper; its relevance to the modeling problem will be shown in a subsequent publication. What we will use instead is the theory of how you *compute* rate-distortion functions. In particular, by choosing a uniform distribution over $\{x_1, \cdots, x_N\}$ and by not requiring that these are unique, we are in effect setting the source distribution to be the *empirical* distribution observed in the training data.

## 3   THE MAIN MATHEMATICAL RESULTS

In latent variable modeling, the correct choice for the distortion function turns out to be

$$d(x,z) \triangleq -\log \ell(x|z).$$

The fundamental reason for this will be revealed in the statement of Theorem 1 below. Computing a rate-distortion function starts by noting that the constraint in the optimization (8) defining the rate-distortion function can be eliminated by writing down the Lagrangian:

$$\min_{Q_{Z|X}} \left\{ I(X;Z) - \alpha E_{X,Z} \log \ell(X|Z) \right\}$$

Our first result connects the prior optimization problem (2) with the optimization of the Lagrangian:

**Theorem 1 (prior optimization is an instance of rate-distortion)** *For any $\alpha > 0$,*

$$\min_{p(z)} -\frac{1}{N} \sum_{i=1}^{N} \log \int p(z)\ell(x_i|z)^\alpha dz = \min_{Q_{Z|X}} \left\{ I(X;Z) - \alpha E_{X,Z} \log \ell(X|Z) \right\} \tag{9}$$

The two central actors in this result $p(z)$, $Q_{Z|X}$ both have very significant roles in both latent variable modeling and rate-distortion theory. The prior $p(z)$ is known as the "output marginal" in rate-distortion theory, as it is related to the distribution of the compressed output in a lossy compression system. On the other hand, $Q_{Z|X}$, which is called the "test channel" in rate-distortion theory, is the conditional distribution that one uses when optimizing the famous Evidence Lower Bound (ELBO) which is an *upper bound* to the negative log likelihood (in contrast, (4) is a *lower bound* on the same). In the context of variational autoencoders, which is a form of latent variable modeling, $Q_{Z|X}$ is also called the "stochastic encoder". An optimal prior according to (2) is also an optimal output distribution for lossy compression, and conversely, from an optimal test channel in lossy compression one can derive an optimal prior for modeling.

### 3.1   PROOF OF THEOREM 1

The proof consists on proving "less than or equal" first in (9) and then "more than or equal". We start with the following lemma which is a simple generalization of the Evidence Lower Bound (ELBO). It can be proved through a straightforward application of Jensen's inequality.

**Lemma 1** *For any conditional distribution $Q(z|x)$, $p(z)$, $\ell(x|z)$, and for any $x$, and $\alpha > 0$,*

$$\log \int p(z)\ell(x|z)^\alpha dz \quad \geq \quad -\int Q(z|x) \log \frac{Q(z|x)}{p(z)} dz + \alpha \int Q(z|x) \log \ell(x|z) dz \tag{10}$$

Taking the expectation w.r.t. $X$ in (10) (assuming $X$ is uniform over $\{x_1, \cdots, x_N\}$), and through an elementary rearrangement of terms, one derives that for any $\ell(x|z)$, any conditional distribution

$Q(z|x)$, any $p(z)$, and defining $(X, Z)$ to be random variables distributed according to $\frac{1}{N}Q(z|x)$,

$$\min_{p(z)} -E_X \left[\log \int p(z)\ell(X|z)^\alpha dz\right] \leq -E_X \left[\log \int p(z)\ell(X|z)^\alpha dz\right]$$

$$\leq I(X; Z) + D_{\mathrm{KL}}\left(\sum_{i=1}^N \frac{1}{N}Q(z|x_i)\|p(z)\right) + \alpha E_X \left[\int Q(z|X)(-\log \ell(X|z))dz\right] \quad (11)$$

Since we are free to choose $p(z)$ whatever way we want, we set $p(z) \triangleq \sum_{i=1}^N \frac{1}{N}Q(z|x_i)$ thus eliminating the divergence term in (11). Since this is true for any conditional distribution $Q(z|x)$, we can take the infimum in the right hand side, proving "less than or equal". To prove the other direction, let $p(z)$ be any distribution and define

$$Q(z|x) \triangleq \frac{p(z)\ell(x|z)^\alpha}{\int p(z)\ell(x|z)^\alpha dz} \quad (12)$$

Let $(X, Z)$ be distributed according to $\frac{1}{N}Q(z|x)$. Then

$$\min_{Q(z|x)} \{I(X; Z) + \alpha E_{X,Z}\left[-\log \ell(X|Z)\right]\} \leq I(X; Z) + \alpha E_{X,Z}\left[-\log \ell(X|Z)\right]$$

$$= -D_{\mathrm{KL}}\left(\sum_{i=1}^N \frac{Q(z|x_i)}{N}\|p(z)\right) - E_X \left[\log \int p(z)\ell(X|z)^\alpha dz\right] \leq -E_X \left[\log \int p(z)\ell(X|z)^\alpha dz\right]$$

Since this is true for any distribution $p(z)$, we can take the minimum in the right hand side, completing the proof.

### 3.2 LOWER BOUND ON THE NEGATIVE LOG LIKELIHOOD: OPTIMIZING PRIORS

The goal in this section is to derive the lower bound (4). Theorem 1 suggests that the problem of lower bounding negative log likelihood in a latent variable modeling problem may be related to the problem of lower bounding a rate-distortion function. This is true - mathematical tricks in the latter can be adapted to the former. The twist is that in information theory, these lower bounds apply to (8), where the object being optimized is the test channel $Q_{Z|X}$ whereas in latent variable modeling we want them to apply directly to (1), where the object being optimized is the prior $p(z)$.

The beginning point is an elementary result, inspired by the arguments in Theorem 2.5.3 of Berger (1971), which can be proved using the inequality $\log \frac{1}{u} \geq 1 - u$:

**Lemma 2** *For any $p(z)$, and $\nu(x) > 0$*

$$-\log \int p(z)\ell(x|z)^\alpha dz \geq -\log \nu(x) + 1 - \int p(z)\frac{\ell(x|z)^\alpha}{\nu(x)}dz.$$

Taking the expectation with respect to $X$ in Lemma 2 we obtain

$$-E_X \log \int p(z)\ell(X|z)^\alpha dz \geq -E_X \log \nu(X) + 1 - \int p(z)E_X \left[\frac{\ell(X|z)^\alpha}{\nu(X)}\right]dz$$

For *any* $\tau(x) > 0$, if you substitute $\nu(x) = \tau(x) \int p(z)E_X \left[\frac{\ell(X|z)^\alpha}{\tau(X)}\right]dz$ then we obtain

$$-E_X \left[\log \int p(z)\ell(X|z)^\alpha dz\right] \geq -E_X \left[\log \tau(X)\right] - \log \left(\int p(z)E_X \left[\frac{\ell(X|z)^\alpha}{\tau(X)}\right]dz\right)$$

We now eliminate any dependence on $p(z)$ on the right hand side by taking the sup, and then take the min over $p(z)$ on both sides, obtaining that for any $\tau(x) > 0$,

$$\min_{p(z)} -E_X \left[\log \int p(z)\ell(X|z)^\alpha dz\right] \geq -E_X \left[\log \tau(X)\right] - \sup_z \log \left(E_X \left[\frac{\ell(X|z)^\alpha}{\tau(X)}\right]dz\right)$$

The $p(z)$ we eliminated refers to the "optimal" prior, which we don't know (hence why we want to eliminate it). We now bring the "suboptimal" prior: since the above is true for any $\tau(x) > 0$, set $\tau(x) = p(x)$, where $p(x)$ is the latent variable model (1) for a given $(p(z), \ell(x|z))$. Note now that the expression on the left is a log likelihood only for $\alpha = 1$, so choosing this value and assuming that $X$ is distributed uniformly over the training data $\{x_1, \cdots, x_N\}$, then we obtain

**Theorem 2 (information theoretic lower bound on negative log likelihood)**

$$\min_{p(z)} -\frac{1}{N}\sum_{i=1}^{N}\log\int p(z)\ell(x_i|z)dz \geq -\frac{1}{N}\sum_{i=1}^{N}\log p(x_i) - \sup_{z}\log\left(\frac{1}{N}\sum_{i=1}^{N}\frac{\ell(x_i|z)}{p(x_i)}\right). \qquad (13)$$

This bound is *sharp*. If you plug in the $p(z)$ that attains the $\min$, then the $\sup$ is exactly zero, and conversely, if $p(z)$ does not attain the $\min$, then there will be slackness in the bound. It's sharpness can be argued in a parallel manner to the original rate-distortion theory setting. In particular, by examining the variational problem defining the latent variable modeling problem, we can deduce that an optimal $p(z)$ must satisfy

$$\frac{1}{N}\sum_{i=1}^{N}\frac{\ell(x_i|z)}{\int p(z)\ell(x_i|z)dz}\left\{\begin{array}{l}= 1 \text{ if } p(z) > 0 \\ \leq 1 \text{ if } p(z) = 0\end{array}\right.$$

Thus for an optimal $p(z)$ the $\sup$ in Theorem 2 is exactly zero and the lower bound matches exactly with the trivial upper bound (3).

The reader may be wondering: what if we don't know $p(x_i)$ exactly? We reinforce that $p(x_i)$ represents not the *true model*, but rather the proposed model for the data. Still, when performing variational inference, the most typical result is that we know how to evaluate exactly a *lower bound* to $p(x_i)$ (this is true when we are using the Evidence Lower Bound or importance sampling (Burda et al., 2016)). In that case, you obtain an *upper bound* to the gap $\sup_z c(z)$ (see (6) for the definition of $c(z)$). If the statistician wants to get a tighter bound, she can simply increase the number of samples in the importance sampling approach; by Theorem 1 of Burda et al. (2016) we know that this methodology will approximate $p(x_i)$ arbitrarily closely as the number of samples grows.

### 3.3 OPTIMIZING THE LIKELIHOOD FUNCTION FOR A FIXED PRIOR.

A very common assumption is to fix the prior to a simple distribution, for example, a unit variance, zero mean Gaussian random vector, and to focus all work on designing a good likelihood function $\ell(x|z)$. Can rate-distortion theory, which we have only shown relevant to the problem optimizing a prior, say anything about this setting?

We assume that $\mathcal{Z}$ is an Euclidean space in some dimension $k$. Assume that we start with a latent variable model $(p(z), \ell(x|z))$, and then we notice that there is a better choice $\hat{p}(z)$ for the same $\ell(x|z)$, in the sense that

$$-\frac{1}{N}\sum_{i=1}^{N}\log\int \hat{p}(z)\ell(x_i|z)dz < -\frac{1}{N}\sum_{i=1}^{N}\log\int p(z)\ell(x_i|z)dz \qquad (14)$$

Further assume that there is a function $g : \mathcal{Z} \rightarrow \mathcal{Z}$ with the property that if $Z$ is a random variable distributed as $p(z)$, then $Y = g(Z)$ is a random variable distributed as $\hat{p}(y)$. For example, $p(z)$ may describe a unit variance memoryless Gaussian vector and $\hat{p}(y)$ may describe a Gaussian vector with a given correlation matrix; then it is known that there is a linear mapping $g(z)$ such that $Y$ has distribution $\hat{p}(y)$. Another example is when both $p(z)$ and $\hat{p}(z)$ are *product* distributions $p(z) = \Pi_{i=1}^{k}p_i(z)$, $\hat{p}(z) = \Pi_{i=1}^{k}\hat{p}_i(z)$, where each of the components $p_i(z)$, $\hat{p}_i(z)$ are continuous distributions. Then from the probability integral transform and the inverse probability integral transform we can deduce that such a $g(z)$ exists. We do point out that when $p(z)$ is discrete (for example, when $\mathcal{Z}$ is finite), or when it is a mixture of discrete and continuous components, then such $g(z)$ in general cannot be guaranteed to exist.

We regard $p$, $\hat{p}$ as probability measures used to measure sets from the $\sigma$-algebra $\sigma(\mathcal{Z})$. In the Appendix, we prove that if $g(z)$ is a measurable function $g : (\mathcal{Z}, \sigma(\mathcal{Z})) \rightarrow (\mathcal{Z}, \sigma(\mathcal{Z}))$ the two following Lebesgue integrals are identical:

$$\int \ell(x_i|g(z))dp(z) = \int \ell(x_i|y)d\hat{p}(y) \qquad (15)$$

The last integral in (15) is identical to $i$th integral in the left hand side of (14). Therefore, one can define a new $\hat{\ell}(x|z) = \ell(x|g(z))$, and the negative log likelihood of the latent variable model

$(p(z), \hat{\ell}(x|z))$ will be identical to that of $(\hat{p}(z), \ell(x|z))$. A key consequence is that if the lower bound in Theorem 2) shows signs of slackness, then it *automatically implies* that the likelihood function admits improvement for a fixed prior. It is important to note that this is only *one possible type* of modification to a likelihood function. Thus if rate-distortion theory predicts that a prior cannot be improved any more, then this does not imply that the likelihood function cannot be improved - it only means that there is a specific class of improvements that are ruled out.

## 4 EXPERIMENTAL VALIDATION

The purpose of this section is to answer the question: are the theoretical results introduced in this article of practical consequence? The way we intend to answer this question is to pose a hypothetical situation involving a specific image modeling problem where there has been a significant amount of innovation in the design of priors and likelihood functions, and then to see if our lower bound, or quantities motivated by the lower bound, can be used, without the benefit of hindsight to help guide the modeling task.

We stress that it makes little sense to choose an "optimal" likelihood function or "optimal" prior if the resulting model overfits the training data. We have taken specific steps to ensure that we do not overfit. The methodology that we follow, borrowed from (Tomczak & Welling, 2018), involves training a model with checkpoints, which store the best model found from the standpoint of its performance on a validation data set. The training is allowed to go on even as the validation set performance degrades, but only up to a certain number of epochs (50), during which an even better model may be found or not. If a better model is not found within the given number of epochs, then the training is terminated early, and the model used is the best model as per the checkpoints. We then apply our results on that model. Thus, if it turns out that our mathematical results suggest that it is difficult to improve the model, then we have the best of both worlds - a model that is believed to not overfit while simultaneously is close to the best log likelihood one can expect (when optimizing a prior) or a model that cannot be improved much more in a specific way (when improving the likelihood function).

The main analysis tool revolves around the quantity $c(z)$ defined as $c(z) \triangleq \frac{1}{N} \sum_{i=1}^{N} \frac{\ell(x_i|z)}{p(x_i)}$. We know that if $\sup_z \log c(z) > 0$, then it is possible to improve the negative log likelihood of a model by either changing the prior while keeping the likelihood function fixed, or by changing the likelihood while keeping the prior fixed, the scope of improvement being identical in both cases, and upper bounded by $\sup_z \log c(z)$ nats. We also know that if $\sup_z \log c(z) = 0$ then the prior cannot be improved any more (for the given likelihood function), but the likelihood function may still be improved.

Thus $\sup_z \log c(z)$ is in principle an attractive quantity to help guide the modeling task. If the latent space $\mathcal{Z}$ is finite, as it is done for example with discrete variational autoencoders (Rolfe, 2017), then it is straightforward to compute this quantity provided $\mathcal{Z}$ is not too large. However if $\mathcal{Z}$ is continuous, in most practical situations computing this quantity won't be possible.

The alternative is to choose samples $\{z_1, \cdots, z_m\} \subset \mathcal{Z}$ in some reasonable way, and then to compute some statistic of $\{c(z_i)\}_{i=1}^{m}$. Recall that the individual $c(z)$ elements can be used to improve the prior using the Blahut-Arimoto update rule:

$$\log p(z) \leftarrow \log p(z) + \log c(z)$$

If $\log c(z)$ is close to a zero (infinite) vector, then the update rule would modify the prior very little, and because rate-distortion functions are convex optimization problems, this implies that we are closer to optimality. Inspired on this observation, we will compute the following two statistics:

$$\max_i \{\log c(z_i)\}_{i=1}^{m}, \qquad \widehat{\text{Std}} \{\log c(z_i)\}_{i=1}^{m} \tag{16}$$

which we will call the *glossy* statistics, in reference to the fact that they are supporting a generative modeling process using lossy compression ideas. The core idea is that the magnitude of these statistics are an indicator of the degree of sub optimality of the prior. As discussed previously, sub-optimality of a prior for a fixed likelihood function immediately implies sub-optimality of the likelihood function for a fixed prior. These statistics of will vary depending on how the sampling has been done, and thus can only be used as a qualitative metric of the optimality of the prior.

In Variational Inference it is a common practice to introduce a "helper" distribution $Q(z|x)$ and to optimize the ELBO lower bound of log likelihood $\log p(x) \geq \int Q(z|x) \log \frac{p(z)\ell(x|z)}{Q(z|x)} dz$. Beyond its use as an optimization trick, the distribution $Q(z|x)$, also called the "encoder" in the variational autoencoder setting (Kingma & Welling, 2014), plays an important practical role: it allows us to map a data sample $x$ to a distribution over the latent space.

Note that one wants to find values of $z$ for which $\log c(z)$ is "large". One way to find good such instances is to note that the dependency on $z$ is through $\ell(x|z)$. Additionally note that $Q(z|x)$ in essence is designed to predict for a given $x$ values for $z$ for which $\ell(x|z)$ will be large. Our proposal thus is to set $z_i$ to be the mean of $Q(z|x_i)$ for $i = 1, \cdots, N$ (and thus $m = N$). There are many additional variants of this procedure that will in general result in different estimates for the glossy statistics. We settled on this choice as we believe that the general conclusions that one can draw from this experiment are well supported by this choice.

## 4.1 Experiments and interpretation

Our experimental setup is an extension of the publicly available source code that the authors of the VampPrior article (Tomczak & Welling, 2018) used in their work. We use both a synthetic dataset as well as standard datasets. The purpose of the synthetic dataset is to illustrate how the statistics proposed evolve during training of a model where the true underlying latent variable model is known. For space reasons, the results for the synthetic dataset are included in the Appendix. The data sets that we will use are image modeling data sets are Static MNIST (Larochelle & Murray, 2011), OMNIGLOT (Lake et al., 2015), Caltech 101 Silhouettes (Marlin et al., 2010), Frey Faces (FreyFaces), Histopathology (Tomczak & Welling, 2016) and CIFAR (Krizhevsky, 2009). In order to explore a variety of prior and likelihood function combinations we took advantage of the publicly available source code that the authors of the VampPrior article (Tomczak & Welling, 2018) used in their work, and extended their code to create sample from $\mathcal{Z}$ using the strategy described above, and implemented the computation of the $c(z)$ quantities, both of which are straightforward given that the ELBO optimization gives us $Q(z|x)$ and an easy means to evaluate $c(z)$ for any desired value of $z$.

For the choice of priors, we use the standard zero mean, unit variance Gaussian prior as well as the variational mixture of posteriors prior parametric family from (Tomczak & Welling, 2018) with 500 pseudo inputs for all experiments. Our choices for the autoencoder architectures are a single stochastic layer Variational Autoencoder (VAE) with two hidden layers (300 units each), a two stochastic layer hierarchical Variational Autoencoder (HVAE) described in (Tomczak & Welling, 2018) as well as a PixelHVAE, a variant of PixelCNN (van den Oord et al., 2016) which uses the HVAE idea described earlier. Each of these (VAE, HVAE, PixelHVAE) represent distinct likelihood function classes. We are aware that there are many more choices of priors and models than considered in these experiments; we believe that the main message of the paper is sufficiently conveyed with the restricted choices we have made. In all cases the dimension of the latent vector is 40 for both the first and second stochastic layers. We use Adam (Kingma & Ba, 2015) as the optimization algorithm. The log test likelihood reported is not an ELBO evaluation - it is obtained through importance sampling (Burda et al., 2016). We refer the reader to this Tomczak & Welling (2018) for a precise description of their setup.

We now refer the reader to Table 1. We want the reader to focus on this prototypical setting:

*A modeling expert has chosen a Gaussian unit variance, zero mean prior $p(z)$ (denoted by "standard"), and has decided on a simple likelihood function class (denoted by "VAE"). Can these choices be improved?*

The goal is to answer without actually doing the task of improving the parametric families. This setting is the first row in all of the data sets in Table 1. We first discuss the MNIST experiment (top left in the table). The modeling expert computes the max and Std statistics, and notices that they are relatively large compared to the negative log likelihood that was estimated. Given the discussion in this paper, the expert can conclude that the negative log likelihood can be further improved by either updating the prior or the likelihood function.

A modeling expert may then decide to improve the prior (second row of the table), or improve the likelihood function (third and fifth rows of the table). We see that in all these cases, there are improvements in the negative log likelihood. It would be incorrect to say that this was *predicted* by

Table 1: Glossy statistics for our experiments - test NLL in nats. Lower is better. $L$ denotes the number of stochastic layers.

| likelihood function class | prior class | staticMNIST | | | Omniglot | | |
|---|---|---|---|---|---|---|---|
| | | NLL | glossy max stat | glossy std stat | NLL | glossy max stat | glossy std stat |
| VAE | standard | 88.44 | 15.7 | 34.3 | 107.83 | 18.4 | 27.3 |
| | VampPrior | 85.78 | 12.1 | 37.6 | 107.62 | 19.0 | 25.6 |
| HVAE | standard | 86.21 | 11.8 | 38.3 | 103.40 | 16.3 | 33.3 |
| $(L = 2)$ | VampPrior | 84.96 | 9.8 | 40.1 | 103.78 | 14.6 | 31.6 |
| PixelHVAE | standard | 80.41 | 7.3 | 3.0 | 90.80 | 10.3 | 1.0 |
| $(L = 2)$ | VampPrior | 79.86 | 7.7 | 4.1 | 90.99 | 8.0 | 0.8 |

| likelihood function class | prior class | Caltech 101 | | | Frey Faces | | |
|---|---|---|---|---|---|---|---|
| | | NLL | glossy max stat | glossy std stat | NLL | glossy max stat | glossy std stat |
| VAE | standard | 128.63 | 79.4 | 160.5 | 1808.19 | 89.3 | 331.5 |
| | VampPrior | 127.64 | 63.6 | 143.7 | 1746.33 | 82.3 | 349.1 |
| HVAE | standard | 125.82 | 59.4 | 204.8 | 1798.42 | 170.2 | 347.0 |
| $(L = 2)$ | VampPrior | 121.02 | 47.2 | 210.7 | 1761.82 | 152.0 | 325.5 |
| PixelHVAE | standard | 85.79 | 18.4 | 7.4 | 1687.10 | 45.7 | 55.1 |
| $(L = 2)$ | VampPrior | 86.34 | 20.8 | 7.5 | 1676.04 | 9.4 | 33.8 |

| likelihood function class | prior class | Histopathology | | | cifar10 | | |
|---|---|---|---|---|---|---|---|
| | | NLL | glossy max stat | glossy std stat | NLL | glossy max stat | glossy std stat |
| VAE | standard | 3295.07 | 167.8 | 324.5 | 13812.19 | 198.1 | 1325.3 |
| | VampPrior | 3286.61 | 69.5 | 348.9 | 13817.33 | 628.1 | 1303.9 |
| HVAE | standard | 3149.11 | 127.3 | 475.3 | 13400.96 | 479.3 | 1634.4 |
| $(L = 2)$ | VampPrior | 3127.67 | 92.2 | 484.9 | 13399.95 | 195.6 | 1539.0 |
| PixelHVAE | standard | 2636.78 | -0.0 | 0.0 | 10915.34 | 178.4 | 105.5 |
| $(L = 2)$ | VampPrior | 2625.61 | -0.1 | 0.0 | 10961.67 | 246.7 | 36.3 |

the glossy statistics in the first row; instead what we can say is that this was *allowed* - it can still be the case that the statistics allow an improvement on the negative log likelihood, but a particular enhancement proposal does not result on any improvement. Next notice that the std glossy statistic for the fifth row is much smaller than the in the previous configurations. The suggestion is that improving the negative log likelihood by improving the prior is becoming harder. Indeed, in the sixth row we see that a more complex prior class did not result in an improvement in the negative log likelihood. As discussed previously, this does not mean that the likelihood function class can no longer be improved. It does mean though that a particular class of enhancements to the likelihood function - in particular, transforming the latent variable with an invertible transformation - will likely not be fruitful directions for improvement.

The general pattern described above repeats in other data sets. For example, for Caltech 101 the reduction of the max and Std glossy statistics when PixelHVAE is introduced is even more pronounced than with MNIST, suggesting a similar conclusion more strongly. A result that jumps out though is the PixelHVAE result for Histopathology, which prompted us to take a close look - here the statistics are actually *zero*. It turns out that this is an instance where the likelihood function learned to ignore the latent variable, a phenomenon initially reported by Bowman et al. (2016). It also serves as a cautionary tale: the glossy statistics only tell us whether for a fixed likelihood function one can improve the prior, or whether for a fixed prior, a certain type of changes to the likelihood function will actually be an improvement to the negative log likelihood. If the likelihood function is ignoring the latent variable, any prior would be optimal, and no transformation of that prior would result in a better likelihood function, which is what the statistics report. We stress that the exact numbers being reported matter little - a simple change in the sampling process for producing the $\{z_1, \cdots, z_m\}$ will immediately result in different numbers. However our experimentation with changing the sampling process resulted in essentially the same general conclusion we are deriving above.

Based on these experiments, we claim that the information theoretically motivated quantity $\log c(z)$ and its associated glossy statistics (16) do provide useful guidance in the modeling task, and given how easy they are to estimate, could be regularly checked by a modeling expert to gauge whether their model can be further improved by either improving the prior or the likelihood function.

## 5   Conclusions and future work

The main goal for this article was to argue strongly for the inclusion of rate-distortion theory as key for developing a theory of representation learning. In the article we showed how some classical results in latent variable modeling can be seen as relatively simple consequences of results in rate-distortion function computation, and further argued that these results help in understanding whether prior or likelihood functions can be improved further (the latter with some limitations), demonstrating this with some experimental results in an image modeling problem. There is a large repertoire of tools, algorithms and theoretical results in lossy compression that we believe can be applied in meaningful ways to latent variable modeling. For example, while rate-distortion function computation is an important subject in information theory, the true crown jewel is Shannon's famous source coding theorem; to-date we are not aware of this important result being connected directly to the problem of latent variable modeling. Similarly, rate-distortion theory has evolved since Shannon's original publication to treat multiple sources and sinks; we believe that these are of relevance in more complex modeling tasks. This research will be the subject of future work.

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

## 6 APPENDIX

### 6.1 SUPPORTING PROOF FOR EQUATION (15)

In the paper we argue that if one is able to improve a latent variable model by changing the prior, then the same improvement can be attained by changing the likelihood function instead. In order to prove this, we claimed that if $g(z)$ is a measurable function $g : (\mathcal{Z}, \sigma(\mathcal{Z})) \to (\mathcal{Z}, \sigma(\mathcal{Z}))$ the two following Lebesgue integrals are identical:

$$\int \ell(x_i|g(z))dp(z) = \int \ell(x_i|y)d\hat{p}(y) \tag{17}$$

This can be seen as follows: for a measurable set $B \in \sigma(\mathcal{Z})$, by definition

$$p([g(Z) \in B]) = \hat{p}(B)$$

and therefore setting for any given real $\eta$, $B = [\ell(x_i|Y) > \eta]$

$$\hat{p}([\ell(x_i|Y) > \eta]) = p([g(Z) \in \{y : \ell(x_i|y) > \eta\}]) = p([\ell(x_i|g(Z)) > \eta])$$

One definition of the Lebesgue integral is through the Riemann integral:

$$\int \ell(x_i|g(z))dp(z) = \int_0^\infty p([\ell(x_i|g(Z)) > \eta])d\eta = \int_0^\infty \hat{p}([\ell(x_i|Y) > \eta])d\eta = \int \ell(x_i|y)d\hat{p}(y)$$

This demonstrates (15).

### 6.2 EXPERIMENTS ON A SYNTHETIC DATA SET

The theme of the article is on how we can use information theoretic methods to understand how much more a model can be improved. One way to illustrate this is to create training and test data samples coming from a known "true" latent variable model, and then to showcase the statistics that we are proposing in the context of a training event. As the training goes on, we would hope to see how the statistics "close in" the true test negative log likelihood.

For the synthetic data set, in the true latent variable model we have a discrete latent alphabet $\mathcal{Z} = \{0, 1, \cdots, 9\}$, each denoting an digit that we took from a static binarized MNIST data set (Larochelle & Murray, 2011). We assume that $p(z)$ is the uniform distribution over $\mathcal{Z}$, and we assume that $\ell(x|z)$ is a probability law which adds i.i.d. Bernoulli noise ($p = 0.02$) to each of the binary pixels of the digit associated with latent variable $z$. Following the MNIST data set structure, we created a data set with 50K training, 10K validation and 10K test images by sampling from the latent variable model. The 10K test images have a true negative log likelihood score of 78.82.

We refer the reader to Figure 1 for our experimental results on the synthetic data set. For this experiment, we consider only a single stochastic layer VAE with the standard zero mean, unit variance prior. This figure is a graphical depiction of the upper and lower bounds (3) and (4), applied to the test data set negative log likelihood. The figure shows how the estimate of the lower bound starts to approach the true negative log likelihood as the epochs progress - the gap between the upper and lower bounds is exactly

$$\sup_{z \in \{0,1,\cdots,9\}} \log\left(\frac{1}{N}\sum_{i=1}^N \frac{\ell(x_i|z)}{p(x_i)}\right) \tag{18}$$

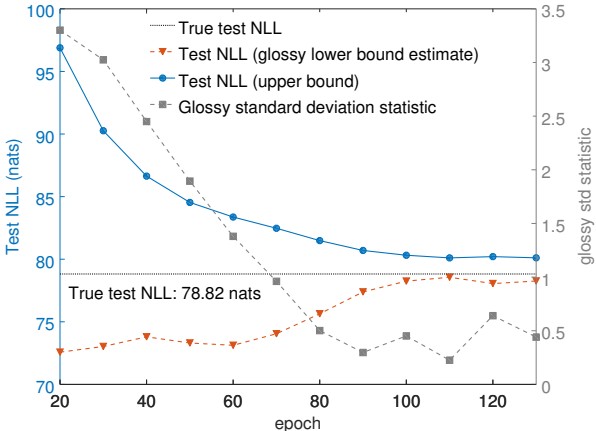

Figure 1: Upper bound and glossy lower bound estimate on negative log likelihood as the training evolves for the synthetic data set. The glossy standard deviation statistic is also overlaid.

which can be computed exactly since the latent variable space $\mathcal{Z}$ is finite. The figure also overlays on a second axis the square root of the glossy variance statistic which we may also call the glossy std statistic.

What we were hoping to show is that the statistics initially show a large potential margin for improvement which eventually shows convergence. As we can see from the Figure, this intuition is confirmed. The gap between upper and lower bounds (18) starts large, but does improve significantly as the training progresses. We also see improvements in the standard deviation statistic, although they are not as dramatic as those in the upper and lower bound gaps - we suspect we had to wait longer to see this effect.

