# OpenReview forum: "Information Theoretic lower bounds on negative log likelihood"
_ICLR.cc/2019/Conference_

### Official Review · AnonReviewer1 · 2018-10-29

**Rating:** 6
**Confidence:** 4

**Review:**

This is an interesting paper that studies the latent variable modeling from an information theoretic perspective. Specifically, the authors argue that the rate-distortion theory for lossy compression provides a natural toolkit for studying latent variable models, and they propose a lower bound (also a gap function) that could be used to assess the goodness of data fitting given a pair of prior distribution over latent factor and a likelihood function. Overall the paper is very well-written, clear to follow, and the authors did a great job in not overclaiming their results.

Several questions follow:
1.  In Eq. (3), why the R.H.S. is an upper bound of the L.H.S.? Under the assumption of (1) should this be equal?
2.  In section 2, "must use at use" -> "must use at least".
3.  Since the mutual information is convex in the conditional distribution Q(Z|X), when considering the Lagrangian, since \alpha is constrained to be positive, should the sign before \alpha be positive instead of negative?
4.  In section 3.3, "An very common" -> "A very common".

To me the most interesting result in this paper is in Thm. 1, Eq. (9), where the authors show that the optimization over the prior in latent variable modeling is exactly equivalent to the optimization of the channel in rate-distortion theory. Following this line the authors propose a gap function that could be used to assess the goodness of a model. One drawback of the current framework is that it only links the optimization of the prior, rather than the likelihood function, to rate-distortion theory, while in practice it is usually the other way around. Although the authors argue in section 3.3 that similar conclusion could be achieved for a family of likelihood functions, the analysis is only possible under the very restrictive (in my personal view) assumption that relies on the existence of a smooth and invertible mapping. This assumption usually does not hold in practice, e.g., the ReLU network, and as a result the analysis here is only of theoretical interest.

The experimental validation basically shows the usefulness of the proposed gap function in assessing the goodness of model fitting in latent variable models. It would be great if there are more direct use of the proposed lower bound, but I appreciate the novelty in this paper on bridging the two subfields.

---

> ### Author Response · Authors · 2018-11-13
> **The fact that if a prior can be improved, then the log likelihood can also be improved holds more generally**
>
>
> In the paper, we argued that results pertaining to the problem of prior optimization were relevant also to the problem of likelihood function optimization (although for the latter problem only weaker statements are possible). In the paper we made an assumption that there existed an invertible mapping $g(z)$ that transformed a random variable $Z$ distributed according to $p(z)$ (the "current" prior) to a random variable with distribution $\hat{p}(z)$ (the "better" prior). Both reviewers 1 and 3 pointed out this as a limitation in the usefulness of the result.
>
> It turns out that the conditions imposed in the original submission were unnecessarily restrictive. The result holds under a very general condition - we only need to assume that $g(z)$ is a measurable function. The claims in the paper have been restated and the proof of this is now included in the Appendix; the proof is a fairly straightforward application of basic concepts in Lebesgue integration. However, we point out if the starting distribution $p(z)$ is discrete, or has some discrete portions (as it is obviously the case when the alphabet \mathcal{Z} is finite), it isn't clear that such a mapping $g(z)$ can be found. In the paper, we do give examples of common continuous distributions where such a mapping is guaranteed to exist (multivariate gaussians and product of continuous distributions).
>
> We now address the individual comments:
>
> 1.  In Eq. (3), why the R.H.S. is an upper bound of the L.H.S.? Under the assumption of (1) should this be equal?
>
> We have overloaded the meaning of $p(z)$ - whenever it is outside of an optimization expression we imply it to be the "current" model that the statistician is trying to improve, and when inside of an optimization expression we think of $p(z)$ as a "free" quantity that one can optimize over. We have clarified in the paper the notation overload. if the reviewer thinks this is still not clear, then we can consider revising the notation in the entire paper.
>
> 3.  Since the mutual information is convex in the conditional distribution Q(Z|X), when considering the Lagrangian, since \alpha is constrained to be positive, should the sign before \alpha be positive instead of negative?
>
> In rate-distortion theory, we want "low rate" (which implies "low" mutual information, which denotes the number of bits used to describe a source sample) and "low distortion". The Lagrangian in rate-distortion thus needs to be setup so that minimizing it promotes reducing mutual information and reducing distortion. In the case of a generative model, whenever $\ell(x|z)$ is high, the meaning is that the data $x$ is well explained by the latent variable $z$ and therefore it corresponds to the notion of "low distortion". Conversely, if $\ell(x|z)$ is very low, then then it means that $x$ and $z$ are mismatched and thus the distortion is high. In the Lagrangian, the logarithm of $\ell(x|z)$ is what appears, but the logarithm is a strictly increasing function. Since $\alpha > 0$, then the correct sign is as written, i.e. I(X;Z) - \alpha \log \ell(x|z).
>
>
> 2. & 4. - suggestions taken.

---

### Official Review · AnonReviewer3 · 2018-11-02
**Novel theoretical analysis, establishing a very formal link between latent-variable models and rate-distortion theory**

**Rating:** 7
**Confidence:** 3

**Review:**

(Please find my response to the rebuttal and updated version in a comment below)
The paper analyses latent-variable modeling from a rate-distortion point-of-view in a novel and interesting fashion, highlighting important fundamental connections. In particular, the paper presents a novel theorem (inspired by how the rate-distortion function is computed) that gives a lower bound on the negative log likelihood. This lower bound allows to quantify by how much a latent-variable model could be improved by either modifying the prior or the likelihood function. The latter is important, since the paper shows a duality between improving one while keeping the other fixed and vice versa. Finally, the paper derives a practical implementation/approximation (founded on solid theoretical analysis) of quantifying the improvement potential of a latent-variable model. These, so called “glossy statistics” are quantitatively analyzed in a set of experiments with different variational autoencoder architectures (various likelihood models and priors) on a number of datasets.

The main contribution of this paper is to provide novel proofs and theoretical analysis that connect latent-variable modeling with rate-distortion on a very fundamental level. While similar attempts have been reported in the recent literature (perhaps a bit more focused on the empirical aspects), the analysis and results in the paper follow a very fundamental treatment of rate-distortion theory and in particular of computation of the rate-distortion function. The central idea underlying rate-distortion, i.e. lossy compression by discarding irrelevant information, seems very suitable as a guiding principle for representation learning. In particular, learning representations that generalize well is essentially another instance of a lossy compression problem. The paper thus addresses an important and timely topic which should be of broad interest to the representation learning community. The paper is well written and mathematically rigorous. I have checked most parts of the proofs, though there still is a chance that I missed something. I am not entirely convinced by the practical impact of the experimental section of the paper (though the experiments are beyond toy-level and I do not doubt the results), but I also believe that this is not the main contribution of the paper, which is rather laying the mathematical groundwork for future work. I vote and argue for accepting the paper for presentation at the conference. My criticism below is aimed at giving some pointers for potentially improving the paper.

1) As the paper acknowledges, there is a risk of overfitting when improving likelihood functions under fixed priors (and vice versa). While the glossy statistics certainly allow making approximate statements of whether the model can be improved further or not, there is no “threshold value” or other guideline that would indicate a modeling expert that they are entering an over-fitting regime if the model-class is further enriched. Therefore, I am not sure about the practical impact of the experiments: the glossy statistics seem to be indicative of the margin for improvements in the negative log-likelihood - but whether all of these improvements are really desirable is unclear. To test this, one might resort to tasks other than generative modeling, such that models that overfit can easily be “spotted” (by degrading test-set performance).

2) Rate-Distortion can be “made more robust” against overfitting by different choices of \alpha (essentially limiting channel capacity). Maybe I am missing something, but shouldn’t the \alpha carry over into the computation of the ratios for c(z)? Was it just assumed to be 1? The same question for Theorem 2 and the equation just above Theorem 2 - does the alpha drop (is it absorbed into the likelihood) or was it set to one? It might be interesting to see how the glossy statistics behave if \alpha is considered a hyper-parameter of the model, e.g. under “low capacity” do the glossy statistics “flatten out” very early?

3) I would have been excited to see how the glossy statistics evolve during training of a model - it would be interesting to show that the statistics initially predict a large margin of improvement that reduces and slowly flattens out as training converges.

4) In the paragraph after Eq. 14: the argument hinges on the possibility of having an invertible (and continuously differentiable) g(z). To me it is not straightforward that a neural network would necessarily implement such a function (particularly the invertibility might be problematic). Is this just a technical condition required for the formal statement, or do you think that this issue could become problematic in practice as well such that the duality between improving prior and likelihood does not hold any longer?

Minor:
5) I think the Alemi et al. reference (first reference) has been published under a different name (Fixing a Broken ELBO) at this years’ ICML.

6) Consider calling the quantity l(x|z) below Eq. 1 “the likelihood of the latent variable given the data” (since the data is given, even though the data is not in the conditional, which is why it is a likelihood function).

7) Rather than using “the KL divergence between”, use “the KL divergence from … to” which nicely reflects its asymmetry.

8) Page 4, last Equation: square brackets for E_X missing

---

> ### Author Response · Authors · 2018-11-13
> **We thank the reviewer for the endorsement!!**
>
> 1) On overfitting - In the experimental setup, we let the training of the models proceed until the log likelihood on a validation data set shows no sign of improvement for a given number of epochs, and then revert back to the best model found during the training. We then calculate the glossy statistics based on such best model. If the glossy statistics indicate that it is becoming harder to improve the model, then we have "the best of both worlds" - a model that is not overfitting that is also becoming harder to improve. If the glossy statistics suggest that it is still possible to improve the model, then we agree that it is possible that the improvement is undesirable as it could lead to overfitting. Still, we believe that the glossy statistics do tell something interesting to the statistician in this case. We believe that a good statistician should be able to spot when it is that her model is likely to overfit, based on experience (observing how a model overfits when the model allows too much freedom), and should be able to interpret the glossy statistics in this case.
>
> We note that in the experiments, we always posted the best model obtained during training in the sense of the procedure involving the validation data set described above (this is, we believe that the models we are posting results for are not overfit). We also believe that the experiments show usefulness of the lgossy statistics.
>
> We have updated the text of the paper to clarify the procedure involving the validation data set.
>
> 2) For the log likelihood expression to be a true log likelihood, we need to set $\alpha = 1$, otherwise the statistics and the lower bound are referring to an expression where the likelihood function is raised to the power of $\alpha != 1$. Since this is a paper devoted to generative models, we chose to simplify to $\alpha = 1$ when we presented the main results - we have now clarified this in the text. Rate-distortion theory of course applies to the entire distortion regime which means it gives results for $\alpha != 1$ as well (and there is a corresponding variation for $c(z)$ in that case). We believe the reviewer is correct in his intuition that by using a different $\alpha != 1$, the training can be made more robust against overfitting, in fact in our experiments (not currently in the paper as submitted) we saw small anecdotal evidence that by setting an $\alpha$ that is close to, but not exactly 1, the resulting model had a slightly improved test data performance. We did not believe the results to be significant enough to be presented as research results at this point. We note that even if we train with $\alpha != 1$, the correct way to interpret the results is to then take the resulting model and plug it into our lower bound and glossy statistics with $\alpha = 1$, so that we can obtain a statement for the straight generative model problem.
>
> 3) We have now added in the appendix what the reviewer is asking for. The effect being sought is indeed present.
>
> 4) We have now eliminated the restriction for the mapping to be invertible and continuously differentiable. The result holds under general conditions - we only require $g(z)$ to be a measurable function. In our initial submission we didn't have enough time to convince ourselves that this was true, but now our Appendix contains a proof of this fact. However, we point out if the starting distribution $p(z)$ is discrete, or has some discrete portions (as it is obviously the case when the alphabet \mathcal{Z} is finite), it isn't clear that such a mapping $g(z)$ can be found. We do give examples in the paper of common settings where this mapping is guaranteed to exist.
>
> 5-8) All suggestions taken.

---

> > ### Comment · AnonReviewer3 · 2018-11-26
> > **Thanks for the clarification and taking into account the reviews**
> >
> > Thank you for clarifying that you are using a validation set to gauge when a model starts over-fitting. I missed that during my first read, but I think it strengthens the experimental results.
> > Also thanks for pointing out the \alpha!=1 case and including it in the main text.
> >
> > I like the additional results in the appendix and I am really happy to see that the authors managed to get rid of the invertibility and continuous differentiability conditions - the two conditions were definitely some pain-points when dealing with deep neural nets.
> > Also (even though it was another reviewer who pointed this out) explicitly clarifying the overloading of p(z) is helpful.

---

### Official Review · AnonReviewer2 · 2018-11-03
**The proposed criterion has has already been examined in some prior works while  little is discussed on the related works.**

**Rating:** 6
**Confidence:** 3

**Review:**

This paper considers the optimization of the prior in the latent variable model and the selection of the likelihood function. The authors propose criteria for these problems based on a lower-bound on the negative log-likelihood, which is derived from rate-distortion theory.

There are some interesting points in the derivation of the proposed quantities and how to compute them while the main criterion c(z) has already been examined in some prior works. Although the results of experiments are promising, they are somewhat weak enough to demonstrate the usefulness of the proposed quantities.

- The note right after Eq.(7) is unclear. It would be nicer to discuss more clearly the property of c(z) about overfitting.

- The derived quantity c(z) in Eq.(6), appearing in the optimality condition (the equation following Eq.(13)), has been pointed out since early times, e.g. in
Lindsay, B. G. (1983). The geometry of mixture likelihoods: A general theory. The Annals of Statistics, 11(1), 86–94.
It was used in the machine learning community too:
Nowozin, S., Bakir, G. (2008). A decoupled approach to exemplar-based unsupervised learning. In Proc. ICML.
, and its connection to rate-distortion theory was pointed out:
Watanabe, K., Ikeda, S. (2014). Entropic risk minimization for nonparametric estimation of mixing distributions. Machine Learning, 99(1), 19–136.
However, little is discussed on these related works.

- Section 2: Shannon's rate-distortion theory is formulated by a general source distribution of X. It would be better to mention that the authors consider the empirical distribution of the data sample as the source distribution.

- The results of experiments only show the potential of glossy statistics in some variational auto-encoder models. Isn't it possible or better to demonstrate its practicality more concretely using small toy models?

Pros:
- nice connection between the optimization of the prior (or likelihood function) and rate-distortion theory

Cons:
- lack of discussion on important related works
- weakness of the experiments

---

> ### Author Response · Authors · 2018-11-13
> **Important related works discussion incorporated**
>
> We thank the reviewer for pointing out very important missing references, which we were not aware of. We are particularly grateful for the pointer to Lindsay' s 1983 paper which we now recognize contains various important results for the problem of prior optimization in latent variable modeling. Beyond what the reviewer points out regarding the quantity $c(z)$ being already highlighted in Lindsey's paper, we also found that our lower bound was also present as well. We also followed the trail of evidence in the other papers that the reviewer pointed to in order to create a more complete picture of prior work in this area.  We have adjusted the paper to reflect this.
>
> Interestingly, Lindsay derived these results without realizing the connection to rate-distortion theoretic results that had been published earlier; even the classic result on the fact that priors with finite support suffice for modeling is a consequence of arguments used in rate-distortion. We still maintain that the relation between rate-distortion theory and latent variable modeling is yet to be fully flushed out and believe that the relatively basic ties we are making to the simplest results in rate-distortion theory will be helpful in constructing this relation.
>
> On the other remarks:
>
> - Section 2: Shannon's rate-distortion theory is formulated by a general source distribution of X. It would be better to mention that the authors consider the empirical distribution of the data sample as the source distribution.
>
> We have updated the paper to explicitly say this.
>
> - The results of experiments only show the potential of glossy statistics in some variational auto-encoder models. Isn't it possible or better to demonstrate its practicality more concretely using small toy models?
>
> We have added in the Appendix results on a smaller toy problem. Although it is still a VAE being trained, the toy problem's training and test data come from a synthetic, known model which allows us to more directly interpret the value of the lower bound and the glossy statistics.

---

> > ### Comment · AnonReviewer2 · 2018-11-23
> > **The revisions surely improved the paper.**
> >
> > The revisions surely improved the paper as the related works are discussed and the added synthetic experiment is helpful for understanding the paper. However, the contribution of this paper is marginal since Lindsay used the same lower bound as the authors discuss and the experiments in the main text are somewhat unsatisfactory, in particular, regarding the treatment of overfitting.

---

> > > ### Author Response · Authors · 2018-11-26
> > > **Comment on overfitting**
> > >
> > > We have revised the text to better reflect how it is that we address the potential problem of overfitting. A copy of this text is pasted below:
> > >
> > > We stress that it makes little sense to choose an “optimal” likelihood function or “optimal” prior if the resulting model overfits the training data. We have taken specific steps to ensure that we do not overfit. The methodology that we follow, borrowed from (Tomczak & Welling, 2018), involves training a model with checkpoints, which store the best model found from the standpoint of its performance on a validation data set. The training is allowed to go on even as the validation set performance degrades, but only up to a certain number of epochs (50), during which an even better model may be found or not. If a better model is not found within the given number of epochs, then the training is terminated early, and the model used is the best model as per the checkpoints. We then apply our results on that model. Thus, if it turns out that our mathematical results suggest that it is difficult to improve the model, then we have the best of both worlds - a model that is believed to not overfit while simultaneously is close to the best log likelihood one can expect (when optimizing a prior) or a model that cannot be improved much more in a specific way (when improving the likelihood function).

---

### Public Comment · ~Xuechen_Li1 · 2018-11-20
**Estimating c(z)**

Hi,

Thanks for an interesting paper that strengthens the connection between latent variable modeling and rate-distortion theory.

More practically, I am interested in the computation of c(z) in the paper, which does not seem to be too obvious unless I am missing something. I believe the p(x_i) terms that appear in the formula for c(z) make exact computation less likely. Could you elaborate on this?

Thanks in advance!

---

> ### Author Response · Authors · 2018-11-21
> **Updated paper to clarify this point**
>
> Thanks for the comment!! The short answer is that most of the time you are estimating an upper bound to $c(z)$, but that through importance sampling ideas, you can approximate $c(z)$ arbitrarily closely, and doing that is not that difficult to do; for example the authors of the VampPrior article in their source code (see the main line option https://github.com/jmtomczak/vae_vampprior/blob/bb6ff3e58036adbea448e82d7bc55593d605b52c/experiment.py#L77) already have a parameter to improve the approximation the log likelihood in a way done *outside* of the main training loop.
>
> Here's the new paragraph that has been added in response:
>
>
> The reader may be wondering: what if we don't know $p(x_i)$ exactly? We reinforce that $p(x_i)$ represents not the \emph{true model}, but rather  the proposed model for the data. Still, when performing variational inference, the most typical result is that we know how to evaluate exactly a \emph{lower bound} to $p(x_i)$ (this is true when we are using the Evidence Lower Bound or importance sampling  \citep{DBLP:journals/corr/BurdaGS15}). In that case, you obtain an \emph{upper bound} to the gap $\sup_{z} c(z)$ (see (\ref{eq:c}) for the definition of $c(z)$). If the statistician wants to get a tighter bound, she can simply increase the number of samples in the importance sampling approach; by Theorem 1 of \citet{DBLP:journals/corr/BurdaGS15} we know that this methodology will approximate $p(x_i)$ arbitrarily closely as the number of samples grows.

---

### Meta-Review · Area_Chair1 · 2018-12-14
**Skirts close to previous work but ultimately novel**

**Confidence:** 3
**Recommendation:** Accept (Poster)

**Metareview:**

Strengths:  This paper gives a detailed treatment of the connections between rate distortion theory and variational lower bounds, culminating in a practical diagnostic tool.  The paper is well-written.

Weaknesses:  Many of the theoretical results existed in older work.

Points of contention:  Most of the discussion was about the novelty of the lower bound.

Consensus:  R3 and R2 both appear to recommend acceptance (R2 in a comment), and have both clearly given the paper detailed thought.